# Release of CD36-associated cell-free mitochondrial DNA and RNA as a hallmark of space environment response

Nailil Husna[1,2], Tatsuya Aiba [3], Shin-Ichiro Fujita[1,7], Yoshika Saito[4], Dai Shiba [3], Takashi Kudo [5,6], Satoru Takahashi [5,6], Satoshi Furukawa [3] & Masafumi Muratani [1,5]

A detailed understanding of how spaceflight affects human health is essential for long-term space exploration. Liquid biopsies allow for minimally-invasive multi-omics assessments that can resolve the molecular heterogeneity of internal tissues. Here, we report initial results from the JAXA Cell-Free Epigenome Study, a liquid biopsy study with six astronauts who resided on the International Space Station (ISS) for more than 120 days. Analysis of plasma cell-free RNA (cfRNA) collected before, during, and after spaceflight confirms previously reported mitochondrial dysregulation in space. Screening with 361 cell surface marker antibodies identifies a mitochondrial DNA-enriched fraction associated with the scavenger receptor CD36. RNA-sequencing of the CD36 fraction reveals tissue-enriched RNA species, suggesting the plasma mitochondrial components originated from various tissues. We compare our plasma cfRNA data to mouse plasma cfRNA data from a previous JAXA mission, which had used on-board artificial gravity, and discover a link between microgravity and the observed mitochondrial responses.

A detailed understanding of how the human body responds to spaceflight will be important to develop countermeasures for long-term space exploration[1]. Without effective solutions for aerospace health ailments, increasingly critical challenges will manifest as humanity ventures into crewed interplanetary travel. Solid biopsies have traditionally served as the gold standard for molecular monitoring of human physiology, but such invasive procedures require specialized equipment and carry an increased risk of infection and regional pain. To circumvent these difficulties, researchers on Earth have started to use blood or fluid samples, which are simpler to perform and require little to no recovery time, allowing for increased temporal resolution of sampling. These minimally invasive "liquid biopsies" permit the extraction of cell-free nucleic acids[2,3], which are

thought to detect changes earlier than conventional protein biomarkers[4] and to better resolve molecular heterogeneity than standard tissue biopsies[5]. Cell-free nucleic acids have emerged as potential space-relevant biomarkers of psychosocial and physical stress[6], aging[7], metabolic disorders[8], and inflammation[9], and have already been successfully utilized in various terrestrial healthcare contexts, including screening for infections[10], diagnosing cancer[11], and monitoring cancer recurrence and therapy[12]. On a similar note, cell-free nucleic acids can infer well-documented spaceflight responses, such as DNA damage[13,14], in distal tissues without the need for invasive tissue biopsies[15], which would be particularly difficult to conduct in orbit.

Longitudinal and full-body molecular profiling using cell-free nucleic acids isolated from liquid biopsies is hence one viable avenue

[1]Department of Genome Biology, Institute of Medicine, University of Tsukuba, Ibaraki 305-8575, Japan. [2]Program in Humanics, University of Tsukuba, Ibaraki 305-8573, Japan. [3]Human Spaceflight Technology Directorate, Japan Aerospace Exploration Agency (JAXA), Ibaraki 305-8505, Japan. [4]Faculty of Medicine, Kyoto University, Kyoto 606-8303, Japan. [5]Transborder Medical Research Center, University of Tsukuba, Ibaraki 305-8575, Japan. [6]Department of Anatomy and Embryology, Institute of Medicine, University of Tsukuba, Ibaraki 305-8575, Japan. [7]Present address: Department of Neurobiology, Northwestern University, Evanston, IL 60201, USA. ✉e-mail: muratani@md.tsukuba.ac.jp

for advancing space healthcare. This approach may grant a detailed understanding of how the human body responds to the space environment and the development of modern countermeasures. Consenting astronauts could participate in lifetime surveillance through annual liquid biopsies, which could reveal the degree of full recovery from spaceflight responses, such as DNA damage and clonal hematopoiesis (CH), the latter of which causes increased lifelong risk of cardiovascular disease and blood cancers[16,17]. Liquid biopsies may be particularly constructive during interplanetary spaceflight missions, in which orbital healthcare autonomy will need to be optimized due to the large distance away from Earth. With the advent of the portable, nanopore-based Oxford Nanopore Technologies MinION, nucleic acids could potentially be extracted and directly sequenced in-flight by crew to monitor their own health in near real time[18–20].

The NASA Twins Study provided a unique opportunity to perform a multi-omics analysis with monozygotic twin astronauts, where one participant resided on the ISS for 340 days, and the other remained on Earth[16]. The past few years have now welcomed the first studies to examine cell-free DNA (cfDNA) in astronauts, using data from the NASA Twins Study[21] and data collected 20 years prior with 14 astronauts on short-term missions[22]. These studies highlighted a theme of mitochondrial stress as a consistent spaceflight phenotype[23], with bulk RNA-sequencing (RNA-seq) of peripheral blood mononuclear cells (PBMCs) revealing mitochondrial RNA (mtRNA) spikes during flight, the levels of which were correlated to time spent on the ISS[24], and plasma analysis revealing cell-free mitochondrial DNA (cf-mtDNA) spikes during flight[21] and in the immediate days after return[22].

Here, we report an independent study with six astronaut participants, where we examined plasma cfRNA profiles that reflect the longitudinal responses of internal tissues to the spaceflight environment. To disentangle microgravity from the various factors of the spaceflight environment, we directly compared our human plasma cfRNA data to mouse plasma cfRNA data from a published JAXA spaceflight mission that incorporated an artificial gravity device installed on the Japanese ISS Kibo Experiment Module[25]. Through this process, minimally invasive approaches were used in both species to gain a comprehensive overview of conserved environmental responses to microgravity, with increased mitochondrial components in plasma being one such prominent response.

## Results

### Plasma cfRNA analysis of six astronauts
Blood samples were collected from six astronauts before, during, and after spaceflight on the ISS. An 11-time point sampling course was designed to capture the pre-flight baseline, the early to late phases of more than 120 days on the ISS, and the early to late phases of readaptation in the first 120 days after flight (Fig. 1a). Total cfRNA and cfDNA were purified from plasma samples and their yields were determined (Supplementary Fig. 1). To verify overall sample quality, real-time PCR analysis was performed to measure the ratio of chromosomal to mitochondrial DNA in total cfDNA. Relative copy numbers of mitochondrial DNA in plasma significantly increased during spaceflight but returned to baseline levels in post-flight samples (Fig. 1b), similar to previous reports[21].

All 66 cfRNA samples were processed for RNA-seq analysis. Data quality was assessed based on visualization of mapped reads and principal component analysis (PCA) plots (Supplementary Figs. 2, 3). Two samples (one pre- and one post-flight) were removed from further analysis due to low sequencing library yield and poor mapping quality. PCA plots indicated variations between the six astronauts in the pre-flight phase. As these variations caused difficulties in defining pre-flight baselines, pairwise comparisons of cfRNA profiles using pre-flight baseline samples were not effective (Supplementary Fig. 4 and Supplementary Data 1). However, RNA profiles showed distinct differences between in-flight and post-flight samples. This allowed us to effectively identify altered RNA species between in-flight and post-flight phases, even with the relatively small number of six astronauts participating in our study (Fig. 1c, d). All time points were consolidated into pre-, in-, and post-flight groups. Statistical analysis was performed to identify 466 differentially represented cfRNAs (DRRs) (ANOVA tool in CLC Genomics Workbench, FDR-adjusted $P$-value < 0.05, |fold change| > 2, |difference| > 50, Supplementary Data 2). Groups of genes with similar fold-change values were identified through scatterplots, where they formed quasi-straight lines that deviated from the 45-degree diagonal (Fig. 1d). Prominent examples included cfRNAs from mitochondrial genes (13 DRRs) and cfRNAs from mucin family genes (8 DRRs), which were increased and decreased respectively during spaceflight (Fig. 1d). Subplots e–j in Fig. 1 show normalized quantification values for individual cfRNA species based on mapped genes throughout the time course. cfRNAs from a subset of mitochondria-encoded genes were consistently elevated from in-flight +5 days, reaching a peak at in-flight +30 days, and decreasing at post-flight +3 days. cfRNAs from mucin gene family members decreased during spaceflight and increased transiently during the post-flight phase.

### Molecular pathways affected by spaceflight
cfRNA provides a unique opportunity to estimate molecular changes occurring in the internal tissues of the human body. To explore the functional categories of altered cfRNAs, a pathway analysis was performed using Metascape 3.5[26]. Consistent with the alteration of mitochondrial RNAs, enrichment of pathways related to reactive oxygen species (ROS) was detected (Fig. 2a). Identification of pathways related to the nervous system, including behavior, regulation of synapse organization, modulation of chemical synaptic transmission, and the neuronal system, indicated that blood sampling successfully monitored molecular responses in nervous tissues. Tissue-specificity analysis further supported the alteration of skeletal muscle and the cerebellum (Fig. 2b). Correlation analysis of the 466 DRRs indicated two major coregulated cfRNA clusters (Supplementary Fig. 5 and Supplementary Data 3). Mitochondrial RNAs were clustered together, but cfRNAs linked to other pathway terms did not show clear segregation.

Time-course profiles of cfRNAs linked to these tissue processes indicated that baseline levels of these RNAs tended to decrease during spaceflight and recover during post-flight phases (Fig. 2c–k). Cerebellum-related genes calcium voltage-gated channel subunit alpha1 A (CACNA1A), HR (HR lysine demethylase and nuclear receptor corepressor), immunoglobulin superfamily member 9B (IGSF9B), and paired box 6 (PAX6) and skeletal muscle-related genes immunoglobulin-like and fibronectin type III domain containing 1 (IGFN1), ryanodine receptor 1 (RYR), and xin actin-binding repeat containing 1 (XIRP1) showed reduction during spaceflight. RNA levels of these genes showed a rapid increase at post-flight +3 days and decreased by post-flight +120 days. Skeletal muscle-related genes MAP3K7CL (MAP3K7 C-terminal like) and SH3BGRL (SH3 domain binding glutamate-rich protein like) showed opposite changes, with increased representation during spaceflight. Similar to mitochondrial RNAs, there were delays in the changes of RNA levels for tissue-specific genes during the transition from pre-flight to in-flight phases, but re-adaptation responses to the ground environment were faster.

cfRNAs from genes involved in the vitamin D receptor signaling pathway, KANK2 (KN motif and ankyrin repeat domains 2), MN1 (MN1 proto-oncogene, transcriptional regulator), RXRA (retinoid X receptor alpha), and TRIM24 (tripartite motif containing 24), included a nuclear receptor and co-regulators. These RNA species were also altered in opposite directions between in-flight and post-flight phases (Fig. 2l–o). Vitamin D plays a central role in inter-organ communication through hormones secreted by the parathyroid gland, which regulates skeletal and muscular health[27]. Changes in cfRNAs derived from these

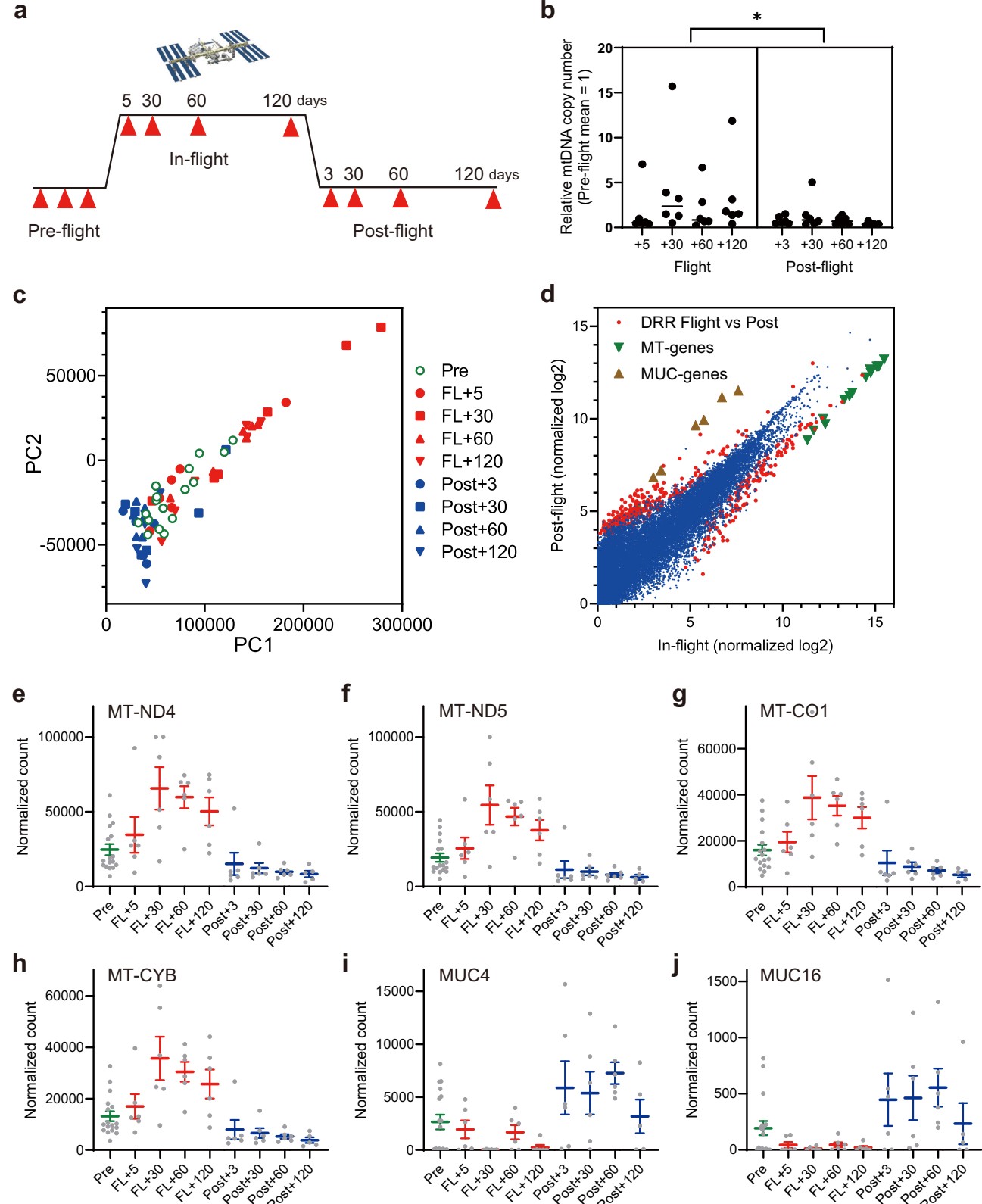

regulatory genes may indicate systemic shifts in mineral and metabolic homeostasis in space.

To further evaluate the feasibility of cfRNA-based internal tissue monitoring, we decided to detect tissue- or cell-type-specific genes in the cfRNA-seq data using more strict criteria. As comprehensive human gene expression databases combined with single-cell RNA-seq, lists of genes with tissue-enriched and tissue-restricted expression were obtained from the Human Protein Atlas (HPA)[28]. Out of the 466 genes derived from our cfRNA analysis, 57 genes were tissue-enriched and 10 of them were detected in a single tissue (Fig. 3a). The majority of cfRNAs from these genes showed relatively low counts, likely due to the minor contribution of cfRNAs originating from each tissue type, but they did show consistent changes throughout the sampling time course (Fig. 3b–f) among the six astronauts. Mapped reads were

**Fig. 1 | Study outline and astronaut cfRNA profiling. a** Time-course sample collection from six astronaut participants. Blood samples were collected at 11 time points before, during, and after spaceflight on the ISS. **b** Real-time PCR quantification of mitochondrial DNA copy number. Quantification values for the MT-ND1 gene were divided by the corresponding globin gene signals. *P*-value = 0.019, Nested one-way ANOVA by GraphPad Prism 9.3.1 software. **c** PCA plot of the RNA-seq quantification results for 64 samples after the removal of two outlier samples. **d** Scatterplot of normalized quantification values for in-flight and post-flight samples. The 64 samples consisted of three groups (pre-, in-, and post-flight). Indicated

are 466 differentially regulated genes (DRRs, red), 13 mitochondrial genes (MT-genes, green), and 8 mucin family genes (MUC-genes, brown). **e–j** Normalized quantification values from RNA-seq for mitochondrial genes and mucin family genes with mean value (solid line) and standard error of the mean (SEM, error bars). SEM values were calculated from $n = 17$ biologically independent samples, including six astronauts for pre-flight, six astronauts for in-flight, and six astronauts for post-flight, except for one Post+120 time point for which one sample was removed due to poor data quality.

---

visually inspected (Fig. 3g–k) as we did not have enough materials to perform PCR-based validation for multiple RNA species. To verify tissue-specificity with an independent database, tissue-specific genes from the HPA were compared to the Genotype-Tissue Expression (GTEx, https://gtexportal.org/) database record (Fig. 3l–p). RNA expressions from the DNM3 (dynamin 3), C1QL3 (complement C1q 3), and AMER2 (APC member recruitment protein 2) genes were highly enriched in the brain and pituitary. RNAs from the KRT2 (keratin 2) and MUC5B (mucin 5B) genes were expressed specifically in the skin and minor salivary glands among the tissues and cell types included in the HPA and GTEx database. These results support that our cfRNA-based liquid biopsy approach allowed for the detection of spaceflight-associated responses in non-blood tissues.

## Isolation of mitochondrial fraction

An increase in plasma cell-free mitochondrial DNA and RNA indicated the presence of extracellular mitochondria (exMT) or the release of cytosolic mitochondrial components into the bloodstream during spaceflight. As unprotected nucleic acids in plasma are expected to be highly unstable, we hypothesized that cfDNA and cfRNA derived from mitochondria may be protected inside extracellular vesicles (EVs). We decided to isolate such EVs based on cell surface antigens for further characterization, as previously described[29]. To identify potential surface markers of EVs containing mitochondrial components, a panel of 361 antibodies was screened for enrichment of mtDNA (Fig. 4a). Antibodies were pooled and mixed with in-flight and post-flight plasma samples, and antibody-bound materials were purified. Real-time PCR was used to evaluate the recovery of mtDNA in each purified material. Through three rounds of antibody screenings, only one antibody, the anti-CD36 antibody, consistently enriched mtDNA (Fig. 4b–d). This enrichment was specific to in-flight plasma samples, consistent with the idea that mitochondrial components released in space are associated with CD36 as a surface molecular marker.

exMT release and inter-organ trafficking have been reported in the context of metabolic stress responses and internal tissue damage associated with ischemia in the heart, adipose tissue, kidney, lung, and brain[30–36]. Cytosolic release of mitochondrial components has been reported in the context of various mitochondrial dysfunctions, which could be subsequently released as EVs during cell death[37]. To investigate the potential tissue origins of the mitochondrial components in plasma, RNA was isolated from the purified fraction and analyzed by RNA-seq. RNAs from 406 genes were enriched in the CD36 fraction (Fig. 4e, CLC Genomics Workbench, Empirical analysis of DGE, FDR-adjusted *P*-value < 0.05, |fold change| > 2, Supplementary Data 4). A tissue-specificity analysis using Database for Annotation, Visualization and Integrated Discovery (DAVID) tools[38,39] revealed that the RNAs in the CD36 fraction isolated from in-flight plasma samples were derived from genes with diverse tissue-specificities. The list of tissues included the brain, skeletal and heart muscles, and relatively minor tissue types, including the retina, choroid plexus, tongue, skin, and neuroendocrine organs, namely, the pituitary, thyroid, and parathyroid glands (Fig. 4f, Supplementary Data 5). Some of these tissue types have not been previously analyzed, even in mouse spaceflight missions. One advantage of plasma fractionation using the anti-CD36 antibody was the successful identification of enriched RNAs that were not detectable

using bulk cfRNA-seq. For example, RNAs encoded by retinal pigment epithelium-specific 65 kDa protein, retinoid isomerohydrolase (RPE65), a component of the visual cycle of the retina that is mutated in hereditary retinal dystrophy[40], and photoreceptor cell components peripherin 2 (PRPH2) and cyclic nucleotide-gated channel subunit beta 1 (CNGB1), both associated with retinitis pigmentosa[41,42], were specifically found in the CD36 fraction.

To estimate the tissue-specificity of the RNAs, expression profiles of RNAs enriched in the CD36 fraction were obtained from the GTEx database. The resulting heatmap revealed subsets of genes highly specific to the cerebellum and brain (Fig. 4g). Other RNAs were also specific to the heart, kidney, liver, muscle, pituitary gland, skin, thyroid gland, and arteries. When the CD36 fraction was purified from post-flight plasma samples, platelet, and megakaryocyte-related RNAs were co-purified as verified by RNA-seq analysis (data not shown). These results indicated that mitochondrial components were released into the plasma during spaceflight as components of CD36-marked EVs, but that this fraction was distinct from EVs derived from platelet degranulation[43,44]. It is also noted that none of the antibodies against other established platelet surface markers included in the antibody panel could enrich mtDNA in our screening[45].

Although CD36 is known as a membrane-bound cell surface protein, it is unclear if the cell-free mtDNA and RNAs were derived from intact exMT or cytosolic mitochondrial components, as is reported in cells under various states of stress[37]. To characterize the CD36 fraction, we examined the status of the mtDNA by sequencing the cfDNA in the CD36 fraction. Fig. 5a shows that the entire mitochondrial genome was contained in the CD36 fraction, which was also supported by the overall similarity of the sequencing read representation between the input and the CD36 fraction. However, it was noticed that the read coverage was higher around the Termination Associated Sequences (TAS) adjacent to the Displacement-loop (D-loop) region. This region contains multiple regulatory cis-elements, including the Conserved Sequence Block (CSB) and promoters for mitochondrial gene transcription[46–48]. Altered sequencing coverage in this region could be due to differences in the regulatory status of the mitochondrial genome. Differences in DNase accessibility may result in shifts in the fragmentation status of purified cfDNA, which affects the efficiency of cfDNA conversion into sequencing libraries (see the "Methods" section for DNA size selection procedures). The D-loop is also reported to show altered DNA accessibility in high-resolution mitochondrial DNA packaging profiles[49].

Mitochondrial RNAs are under transcriptional and post-transcriptional control and are correlated with cell types and disease conditions[50–52]. We performed RNA-seq analysis on the CD36 fraction with extended samples to cover six astronauts over pre-, in-, and post-flight phases. Fig. 5b shows the normalized quantification values for all the mtRNAs from genes encoding core subunits of the oxidative phosphorylation (OXPHOS) system. Clustering these mtRNAs indicated different tendencies between the RNAs from the MT-ATP8, MT-ND3, MT-ND4L, and MT-ND6 genes. Direct plotting of normalized RNA quantification values in Fig. 5c–n showed that ratios among mtRNAs were different between the bulk plasma cfRNA samples and the RNAs purified from the CD36 fraction in a subset of genes. For example, ratios of normalized quantification values between MT-ND4 and MT-

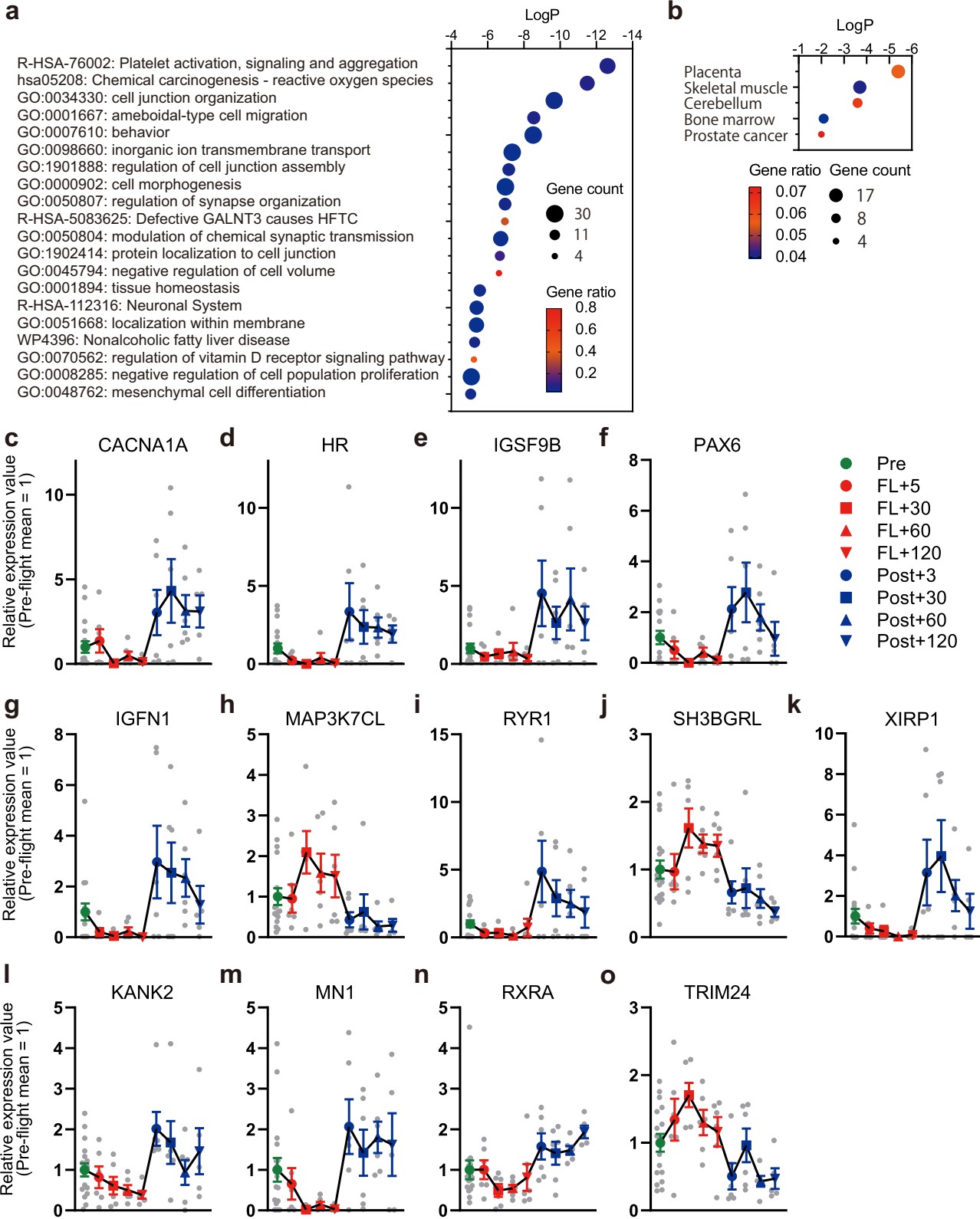

**Fig. 2 | Pathway analysis of DRRs from cfRNA analysis. a** Enriched terms for the 466 DRRs from the cfRNA analysis with LogP, gene count, and gene ratio (the number of genes in each term divided by the number of genes in the same category in the database) based on our Metascape enrichment analysis. **b** Enriched tissue types of the 466 DRRs based on our Metascape PaGenBase analysis. **c**–**k** Relative expression values obtained by RNA-seq for DRRs under neurological and skeletal muscle terms throughout the time course. Mean value (solid line) and −/+ SEM (error bar) from the same biologically independent samples used for Fig. 1e–j are shown. **l**–**o** Relative expression values obtained by RNA-seq for DRRs under vitamin D receptor signaling pathways throughout the time course. Mean values (solid line) and SEM (error bar) were calculated from the same biologically independent samples as (**c**–**k**).

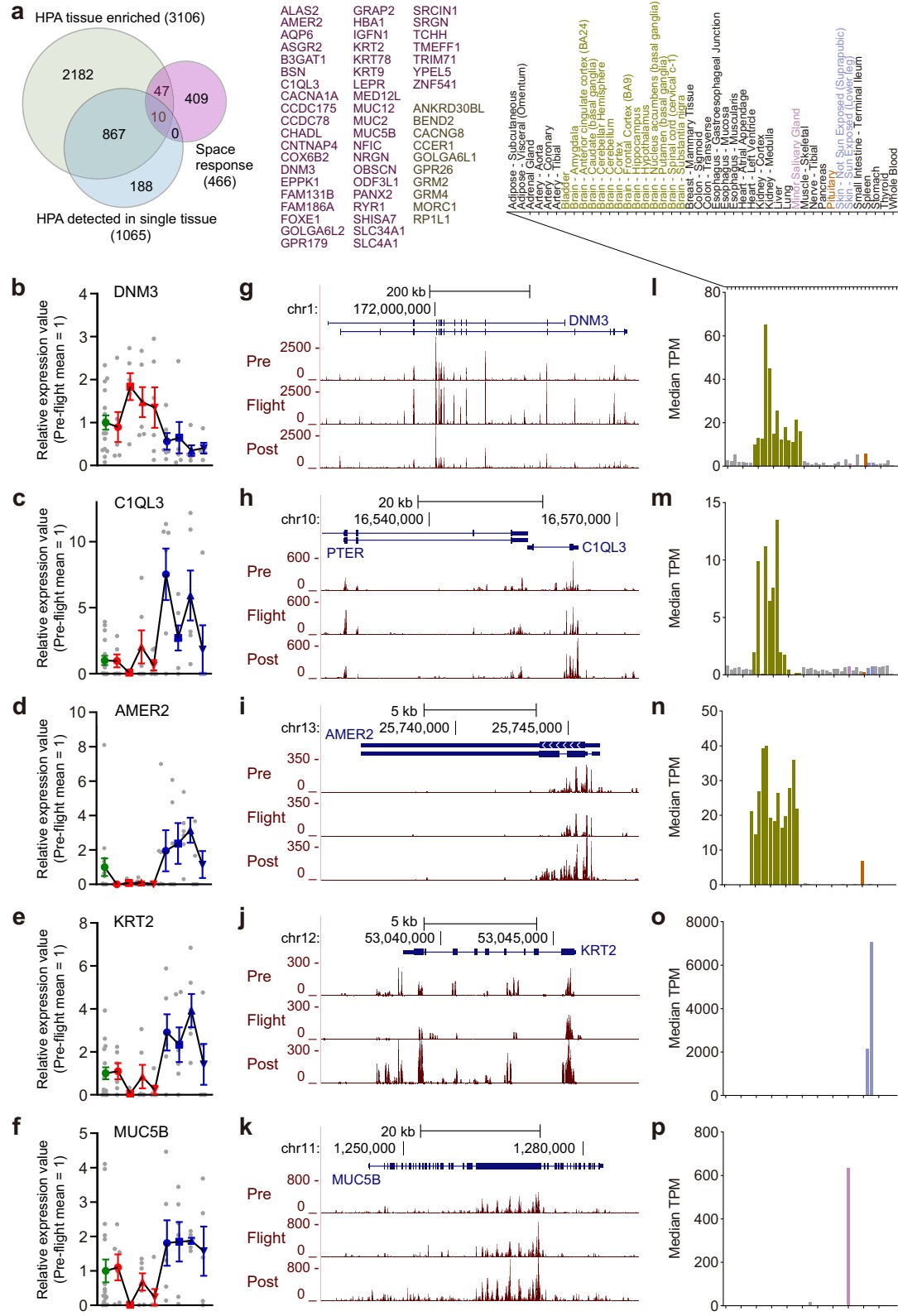

**Fig. 3 | Validation of candidate tissue-specific cfRNAs. a** Venn diagram comparing the 466 space response DRRs and the two gene lists from the HPA database. Gene names for overlapping parts are listed. **b–f** Normalized quantification values from bulk plasma cfRNA-seq for the DNM3, C1QL3, AMER2, KRT2, and MUC5B genes with mean values (solid lines) and SEM (error bars) calculated from the same biologically independent samples used for Fig. 1e–j. **g–k** Visualization of mapped cfRNA-seq reads for the genes shown in panels (**b–f**). Reads from pre-, in-, and post-flight samples were pooled and loaded to the UCSC Genome Browser with representative Ensemble transcript annotations. **l–p** Tissue expression profiles for genes in panels **b–f** were obtained from the GTEx database. The horizontal axis represents the corresponding tissue types listed above the graphs.

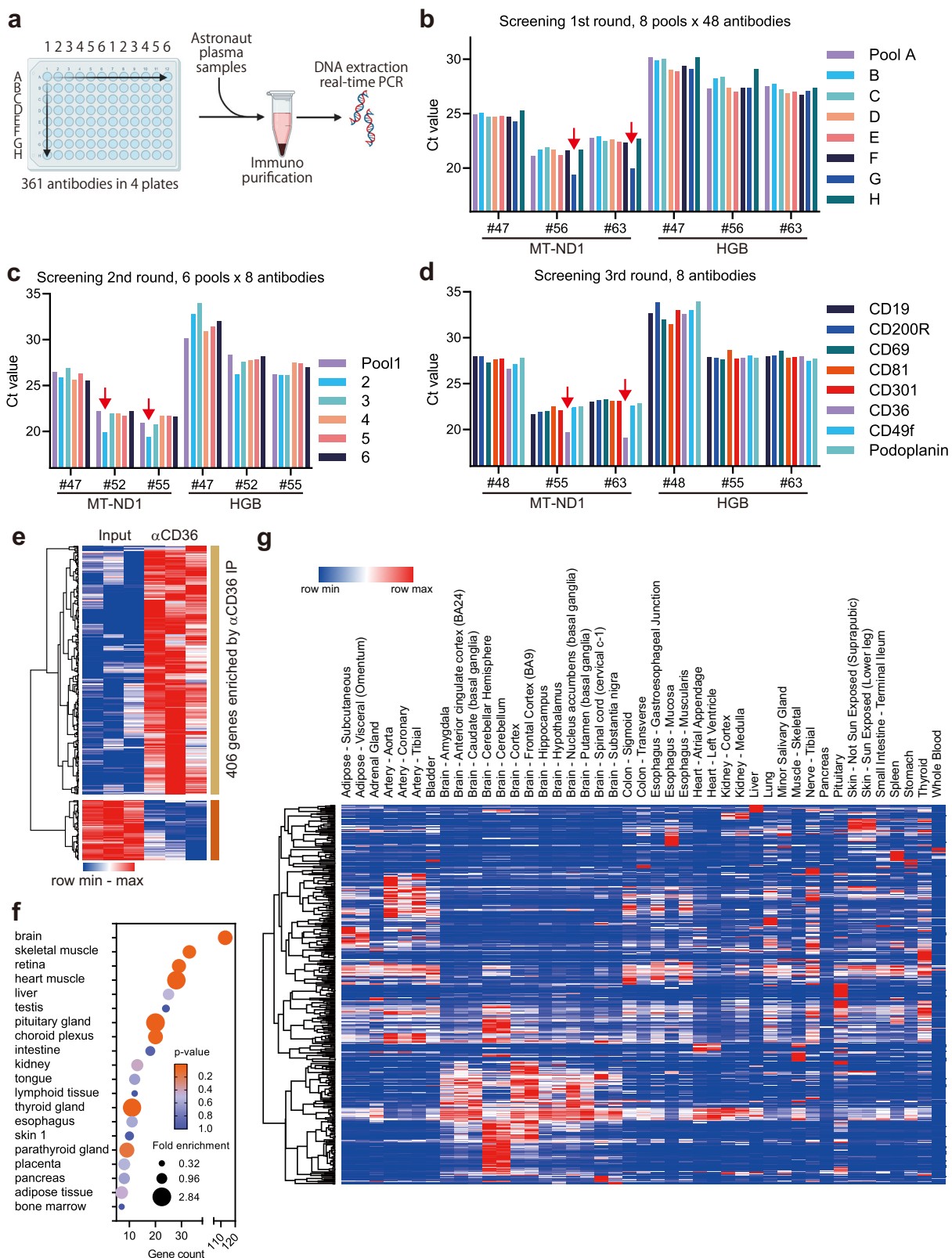

ND5 (Fig. 5l) were comparable. In contrast, ratios between MT-ND4 and MT-ND6 normalized quantification values in the CD36 fraction were 2.57-fold larger than in the bulk plasma cfRNA (Fig. 5d). ND6 has been reported as an important factor in human diseases[52,53], and specific transcriptional and port-transcriptional regulation has been reported[50,51]. These results indicate a specific molecular status of the mitochondrial DNAs and RNAs contained in the CD36 fraction.

## Cross-species comparison of blood samples

Blood is an important sample type that can directly link humans and other species in a minimally invasive fashion. Sample collection in humans is highly limited in space, but time-course sampling from the same astronaut compensates for the small number of participants and creates biological replicates for each subject. The increased sample volume in humans is also advantageous for multi-omics analysis and

**Fig. 4 | Identification and characterization of exMT-enriched CD36 fraction.**
**a** Screening of an antibody panel against cell-surface markers. Antibodies from four 96-well plates were pooled and mixed with in-flight or post-flight astronaut plasma samples. Created with BioRender.com. **b** Purified DNA from each immunopurification was analyzed by real-time PCR using the MT-ND1 and globin (HGB) gene. Red arrows indicate lower PCR cycle threshold (Ct) values, indicating the specific recovery of mtDNA from flight plasma samples (#56 and #63) but not from post-flight samples (#47), through antibody pool G. **c** Second round of antibody screening in which antibodies contained in pool G were subdivided into 6 pools. Pool 2 specifically enriched mtDNA from in-flight samples (#52 and #55) but not from post-flight samples (#47). **d** Third round of antibody screening, which identified the anti-CD36 antibody. **e** Heatmap of differentially purified RNAs between anti-CD36 immunopurification (IP) and input plasma. IP was performed using three different in-flight plasma samples. **f** Tissue enrichment analysis of the 406 CD36-enriched genes against the Human Protein Atlas Tissue Gene Expression Database using DAVID software. **g** RNA expression profile for the 406 CD36-enriched genes obtained from the Genotype-Tissue Expression (GTEx) database.

the fractionation of plasma and other components. Model organism data can provide samples from genetically consistent backgrounds, and time-course sampling can again reduce the number of subjects and hardware needed. In addition to these technical improvements, blood sampling reduces localized pain and infection rates compared to more invasive methods, and hence, it may bring fewer ethical complications compared to traditional techniques[16].

We compared our astronaut blood samples to mouse blood samples from a published 35-day Mouse-Habitat Unit-1 (MHU-1) 2016 JAXA mission to the ISS [25] (Fig. 6a). Ground control (GC) samples had been derived from the same genetic background. Artificial gravity (AG) conditions used rotating platforms to create 1-G gravity controls on the ISS, allowing for the evaluation of gravity effects, while controlling for other flight-related environmental stress factors, such as $CO_2$ concentration, air pressure, isolation, and noise levels.

RNA-seq analysis was performed on the plasma cfRNA mouse data. Similar to our human data, RNAs encoded by mitochondrial genes MT-Cytb (cytochrome b, mitochondrial), MT-Nd1, and MT-Nd4 (mitochondrially encoded NADH dehydrogenase 1 and 4) were upregulated in the MG condition compared to the AG condition (Fig. 6b–d). By pairwise comparisons among GC, AG, and MG groups, 467 DRRs were identified (Fig. 6e, CLC Genomics Workbench, Empirical Analysis of DGE, $P$-value < 0.05, |fc| > 1.5, Supplementary Data 6).

To evaluate common molecular changes between species, the 467 mouse cfRNA DRRs were converted into 444 human homologs using Metascape gene ID conversion. However, only 14 genes (3.1%) were common to the 466 DRRs identified in our human cfRNA analysis. To compare DRRs at the pathway level, enrichment of human disease terms for DRRs between AG vs. MG and GC vs. MG, which are expected to represent the impact of gravity change and the general spaceflight environment respectively, was performed using the DisGeNET database analysis in Metascape. We found that 60 out of 121 (49.6%) AG vs MG (gravity-linked) disease terms were also observed in our human analysis (Fig. 6f). In contrast, 16 out of 91 (17.6%) GC vs MG (general spaceflight-linked) disease terms were observed in our human analysis (Fig. 6f). These results indicated the presence of common disease-related molecular changes as being specifically associated with the microgravity factor of spaceflight. These disease terms contained previously reported space-related health risks, including muscle atrophy and impairments to the visual system, cardiovascular system, and neurological system[54–58] (Fig. 6g).

## Discussion

Consistent with reports from the NASA Twins Study[21], we found a significant increase in relative mitochondrial DNA copy numbers within cfDNA during spaceflight but a return to baseline levels in post-flight samples (Fig. 1b). Another recent study similarly found a significant increase in cell-free mitochondrial DNA in the plasma of 14 astronauts shortly after landing, although the authors only examined three time points (10 days before launch, day of landing, and 3 days after return)[22]. Our results indicated that the samples collected in our study could be a promising resource to further characterize molecular events associated with spaceflight across a wider range and larger number of time points.

Here, we also show prominent changes in the cfRNA profiles of six astronauts undergoing long-duration spaceflight on the ISS. An increase in plasma mitochondrial components was supported by plasma mtDNA copy numbers and RNAs from mitochondrial genes. Screening with antibodies recognizing cell surface markers identified CD36, and the isolated plasma fraction contained RNA species from genes representing a broad range of tissue specificities. These results not only support the practical use of relatively less invasive blood sampling and liquid biopsy analysis to monitor physiological changes in astronauts but also provide insights into previously uncharacterized tissue types and molecular processes in the human body affected by spaceflight.

cfRNA profiles indicated two distinct modes of molecular changes associated with spaceflight. One group of cfRNAs was represented by mtRNAs which increased in plasma during spaceflight. Another group of cfRNAs was represented by RNAs encoded by mucin genes which decreased during spaceflight. As DNA and RNA molecules exposed to plasma are thought to be highly unstable due to the nuclease activity, we speculated cell-free mtDNAs and mtRNAs in plasma may exist as contents of EVs. Through screening a panel of antibodies, we identified anti-CD36 antibodies that enriched cf-mtDNA from spaceflight plasma samples. In this study, we could not obtain conclusive results to determine whether plasma mitochondrial components represented intact exMTs or cytosolic mitochondrial DNAs and RNAs included in EVs. exMT release into plasma under metabolic stress and tissue injury has been proposed as a defensive mechanism, whereby damaged mitochondria are ejected to protect host cells[32,59]. Mitochondrial DNA released into the cytosol is posited as one of the damage-associated molecular patterns (DAMPs) that activate innate immune pathways[60]. Multiple studies suggest inter-cellular trafficking of damaged mitochondria and ROS as a signal to induce resistance against metabolic stress in distal cells and organs[31,61]. Our results support that a broad range of tissues release mitochondrial components into the plasma during spaceflight, which may indicate systemic metabolic stress responses in the human body under microgravity. Although the mechanisms and biological significance are unclear, mitochondrial components and tissue-specific RNAs could serve as candidate biomarkers to evaluate internal tissue responses in space, and potentially shed light on recent proposals that mitochondrial dysregulation is a central feature that accelerates spaceflight health risks[23].

CD36 has been well studied as a scavenger receptor expressed on various cell types. Although a direct link between CD36-expressing EVs and the release of mitochondrial components has not been reported, CD36 EVs have been described in the context of tissue injury and stress responses[62]. The release of CD36-expressing EVs from endothelial cells and skeletal muscles has been reported[63,64]. Therefore, it is reasonable to infer that the CD36 fractions isolated from spaceflight plasma samples represent CD36 EVs that contain mtRNA and other RNAs derived from the nuclear genomes in their corresponding cells of origin. Our DNA and RNA-seq analysis of the CD36 fraction indicated qualitative differences in the mtDNAs and mtRNAs between the bulk plasma and CD36 fraction samples. Given that transcriptional and post-transcriptional regulation of mitochondrial gene expression has been described in the context of tissue-specificity and disease conditions[50–52], further characterization of the CD36 fraction may

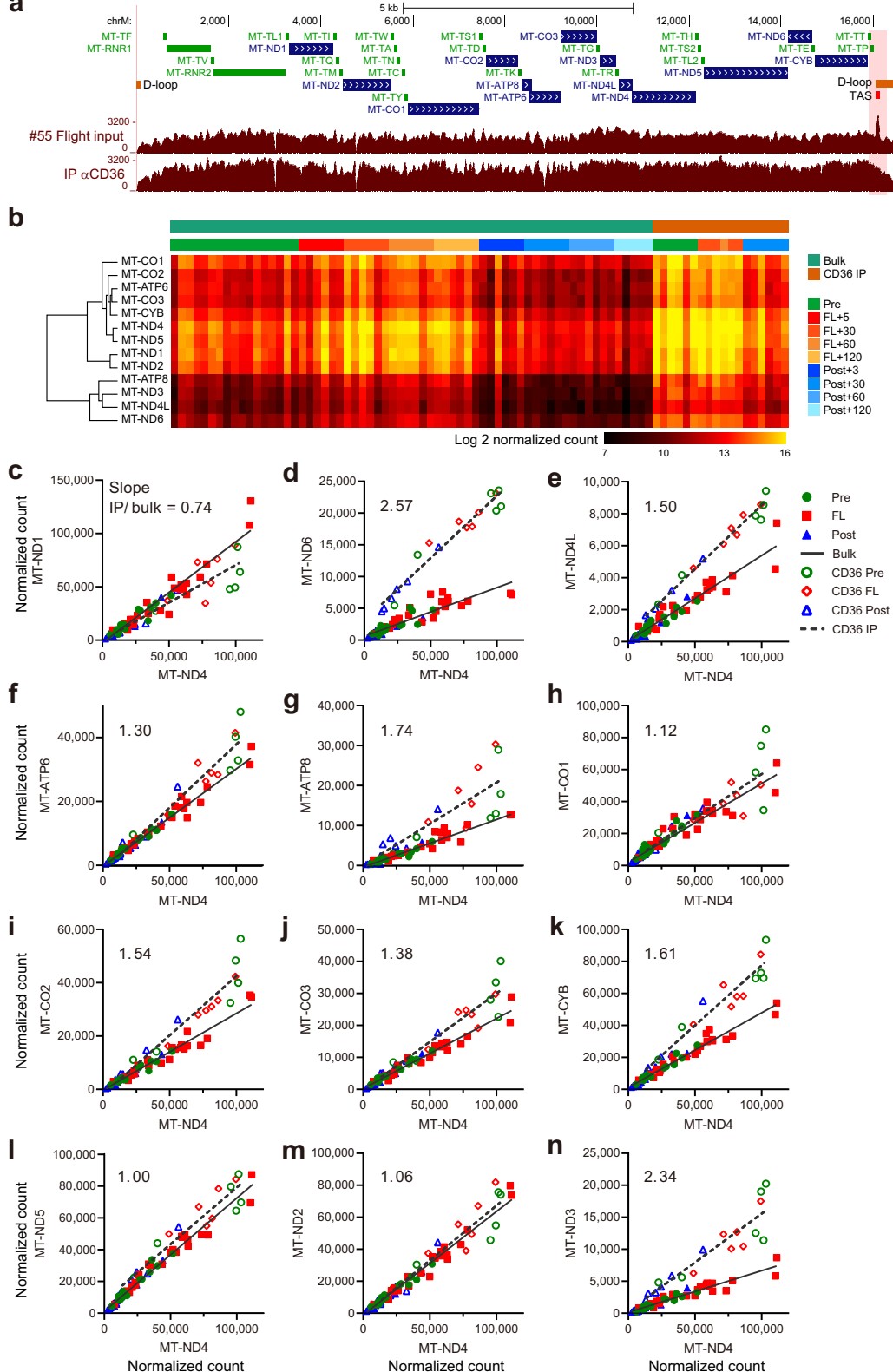

**Fig. 5 | mtDNA and mtRNA profiles in CD36 fraction. a** UCSC Genome Browser view of DNA-seq reads mapped to the mitochondrial genome in bulk plasma cfDNA (input) and DNA purified from the CD36 plasma fraction (IP αCD36). Note that the read coverage peak is seen in the input profile as overlapping with the TAS region of the D-loop, but not in the CD36 fraction. Profiles from sample #55 (Fig. 4d) are shown, and the read coverage peak specific for the input sample was also confirmed for sample #63 input/IP pair (not shown). **b** Heatmap showing normalized

quantification values for mtRNAs in the RNA-seq results for bulk plasma cfRNAs and RNAs purified from the CD36 fraction. An additional set of RNA-seq was performed for the CD36 fraction to cover six astronauts for pre-, in- and post-flight phases, and results were combined with the bulk cfRNA-seq results used for the Fig. 1 analysis. **c–n** Plots showing normalized counts for mtRNAs in plasma cfRNA-seq (bulk) and CD36 fractions (IP). Trend lines were created using GraphPad PRISM 9.3.1, and ratios of the slopes between IP and bulk samples are shown in each panel.

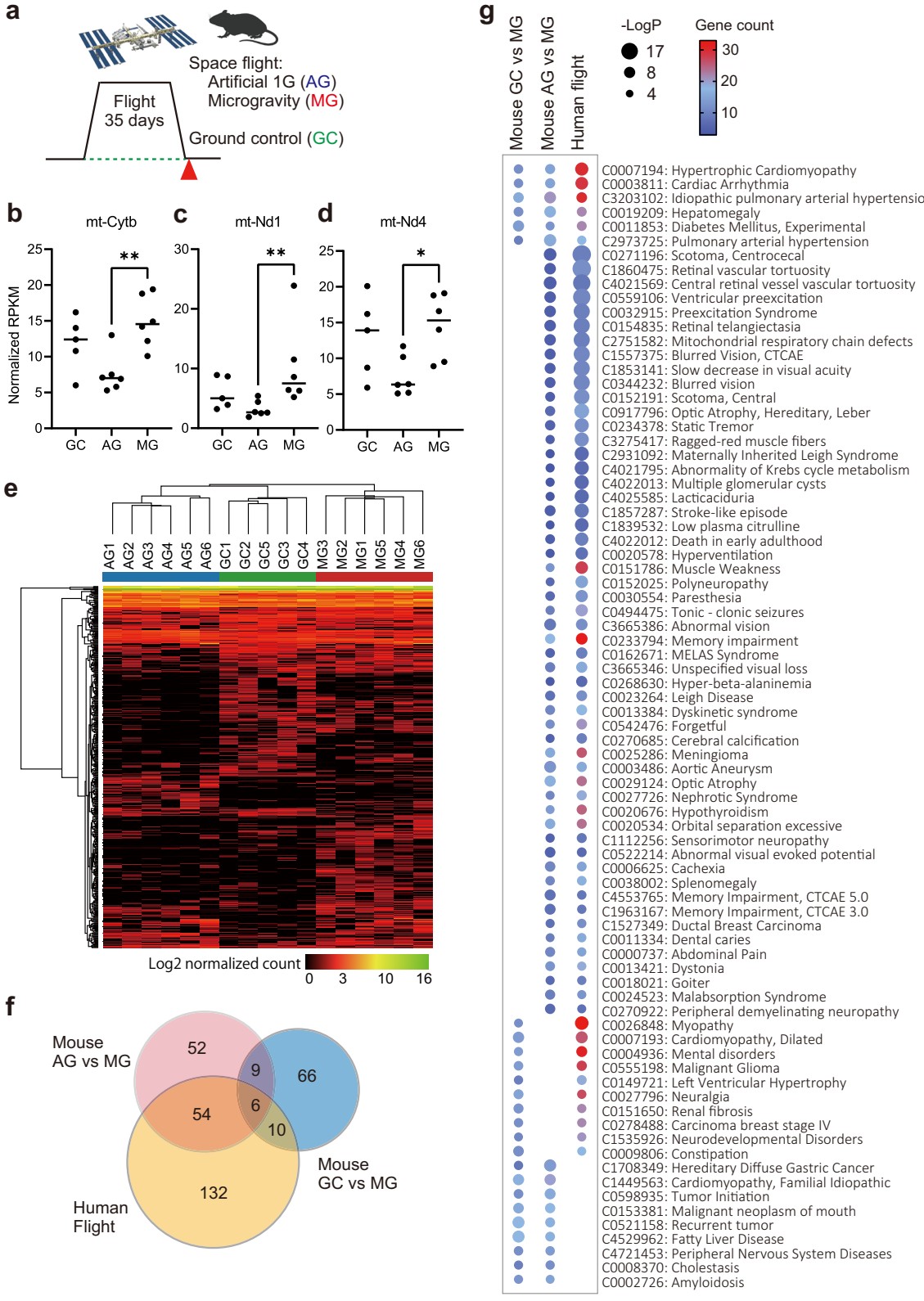

**Fig. 6 | Comparing our human plasma cfRNA with mouse plasma cfRNA from a spaceflight mission. a** Sampling scheme of the MHU-1 spaceflight mission. The 35-day mission in 2016 included six spaceflights (microgravity, MG), six artificial 1-G in-flight controls (AG), and six ground control (GC) mice. Created with BioRender.com. **b**–**d** Mouse cfRNA sequencing results for mitochondrial genes with normalized RPKM (Reads Per Kilobase of transcript per Million reads mapped) values. Solid lines show mean value derived from $n = 6$ (for AG and MG) and $n = 5$ (for GC) biologically independent samples. Results of Mann–Whitney test are shown by ** for two-tailed exact $P$-value < 0.01 and * for <0.05. **e** Heatmap of the 418 DRRs between the GC, AG, and MG conditions. GC samples were collected only from five mice due to a technical problem in blood sampling. **f** Venn diagram showing overlapping DisGeNET terms among DRRs in the AG vs. MG comparison, GC vs. MG comparison, and human cfRNA analysis. **g** Comprehensive list of over-lapping DisGeNET terms from the panel **g** Venn diagram.

provide clues to understanding spaceflight-associated mitochondrial responses.

Tissue-specific RNAs identified space environmental responses in previously not well-characterized tissue types. The pituitary gland, thyroid gland, and parathyroid gland are thought to play central roles in the maintenance and alteration of homeostasis in space, but their small tissue sizes typically prevent their molecular analysis in animal models. These tissues are also inaccessible by direct sampling. It should be noted that the changes in the cfRNA profiles may represent altered tissue contributions as well as RNA expression changes in internal tissues, but our current study has limitations in distinguishing these possibilities.

cfRNAs from mucin family genes were previously not well-characterized in the context of spaceflight response. The mucin genes detected in our study revealed a broad range of tissue-specificities, including the highly tissue-restricted MUC5B (Fig. 3p). Mucins are known to play protective roles in epithelial tissues[65]. Gravity imposes mechanical loading on the entire body, and mucin is expected to be one of the critical components of biolubricants[66]. Therefore, the changes in mucins that we observed may be related to gravity responses. Mucins are also important for functioning as barriers against external pathogens and for protecting tissues against dehydration and adhesion. Our results could highlight the significance of mucins in the maintenance of tissue integrity, and in the controlling of immunological, physical anti-friction, and anti-adhesion functions, through the maintenance of surface tension under the altered gravity environment.

It is unclear if the two modes of cfRNA responses to spaceflight, an increase in mitochondrial components in plasma cfDNA and cfRNA and a decrease in other cfRNAs derived from internal tissues, are mechanistically related. It is also interesting that the tissue-specificity profiles of both groups of cfRNAs are similar, with brain and skeletal muscle as the most prominent tissue types. One of the potential explanations could be altered properties of blood–tissue barrier (BTB) functions. Recent studies reported altered integrity of the blood–brain barrier and blood–retinal barrier in space[67,68]. It is intriguing to note that both studies showed an association between oxidative stress and altered BTB-related molecular processes in space. Murakami et al. have reported seminal studies that directly linked gravity and environmental stressors to BTB regulation and inflammatory responses in the central nervous system (CNS), a process called the gateway reflex[69–71]. Thus, it is possible that the reduced release of cfRNAs from the brain into the plasma could be caused by a reduction in the BTB exchange of cfRNA-containing EVs triggered by microgravity or environmental stresses. As the gateway reflex is linked to pro-inflammatory signals through "IL-6-amplifier activation", potential correlations to neuro-inflammatory responses in the brain and retinal tissues could be the targets for further analysis.

Although the fractionation process in our study was limited by the panel of antibodies used, our results motivate future developments and applications of high-definition liquid biopsies combined with the fractionation of EVs. As more cell-specific surface markers become annotated, the resolution of fractionation will likely improve. Supporting these ideas, we performed another attempt at screening using nuclear-encoded RNAs and identified multiple antibodies that enriched different RNA contents from in-flight and post-flight plasma samples (unpublished results). Either way, improvement in these biomolecular technologies will in turn enable the detection of cell-type-specific gene expression through liquid biopsies, which can be applied for minimally invasive health monitoring and clinical sample analysis on Earth and in space.

As a specific clinical example, the plasma mtDNA-enriched CD36 fraction isolated from in-flight plasma samples contained tissue-specific RNAs indicating eye-related origins (Fig. 4f), and visual diseases were also represented among the gravity-linked terms (Fig. 6g).

These findings indicate a possibility that liquid biopsies could be useful to detect early molecular events associated with retina tissue damage. Spaceflight-associated neuro-ocular syndrome (SANS) is one of the puzzling health issues in astronauts[55]. SANS is present in some astronauts after short and long duration spaceflight[55] and is defined as a constellation of unusual clinical findings, including optic disc edema, globe flattening, choroidal and retinal folds, hyperopic refractive error shifts, and nerve fiber layer infarcts[72]. The cause of SANS remains elusive, but two main hypotheses include elevated intracranial pressure (ICP) and compartmentalization of cerebrospinal fluid (CSF) to the area behind the globe[73]. Our study allowed for the resolution of gene expression in tissues that are traditionally difficult to access in humans, such as the retina and choroid plexus, both of which may be relevant to the study of SANS pathophysiology. Mitochondrial responses and ROS have been reported as characteristic features of neuro-inflammation in CNS and retinal diseases[74]. These changes have also been observed in mouse spaceflight experiments[67,75]. These studies, collectively, could suggest mitochondria-related processes, including the ROS and OXPHOS systems, as potential targets for countermeasure development to prevent health problems associated with spaceflight[76,77]. Overall, this example underlines that minimally invasive liquid biopsies could elucidate new perspectives on the cause of and countermeasures for unique space-related ailments.

This study provided a unique opportunity to compare plasma cfRNA between mammalian species in a minimally invasive manner. The space environment consists of many factors, including microgravity, radiation, confinement, noise, altered air pressure and composition, and stress factors associated with launch and return. The use of on-board artificial 1-G controls in the published MHU-1 mission allowed us to identify gene categories specifically affected by the factor of microgravity, and nearly half (49.6%) of the enriched human disease terms were also observed in our human flight data. These gravity-linked terms were not limited to conventional disuse functions, such as bone and muscle loss, but also included metabolic stress responses, neurological disorders, and importantly, ocular and cardiovascular dysfunctions. With the development of sophisticated liquid biopsy technologies combined with multi-omics approaches, the rapid development of human research in space is expected to provide important insights that benefit human health both on and beyond Earth.

## Methods
### Study registration and ethics
The current human spaceflight study was proposed to and supported by the 2014 International Life Sciences Research Announcements, JAXA, and the National Aeronautics and Space Administration (NASA). Ethics committee approvals were obtained at the University of Tsukuba, Institute of Medicine, Ethics Committee (No. 251, November 27, 2015), JAXA Institutional Review Board for Human Research (JX-IRBA-20–071, August 30, 2016), NASA Institutional Review Board (Pro1995, February 28, 2017), and European Space Agency (ESA) Medical Board (2017_04_09, April 20, 2017). Informed consent was obtained by the personal information manager of the study, and de-identified samples were made available to researchers who performed sample processing and data analysis. The study design and conduct complied with all relevant regulations regarding the use of human study participants and was conducted in accordance with the criteria set by the Declaration of Helsinki. The previously published Mouse Habitat Unit-1 (MHU-1) spaceflight mission is described by Shiba et al.[25].

### Sample collection and transportation
Human blood samples were collected using Vacutainer EDTA-plasma separate gel collection tubes (BD, cat. no. 362788) and frozen at −95 °C (ISS) or −80 °C (ground) after centrifugation for 30 min at $1239 \times g$ (3800 rpm, ISS) or $1600 \times g$ (ground). Frozen samples were transported

to the University of Tsukuba after all samples were collected from the six astronauts. Mouse blood samples were collected from the inferior vena cava under anesthesia, and transferred into a 1.5 ml microfuge tube containing EDTA-2K for a final concentration of 5 mM. Plasma was obtained by centrifugation at 1500×g for 10 min, and collected into a fresh tube. A 100 μl aliquot was frozen and transported to the University of Tsukuba.

### RNA purification

Plasma sample tubes were thawed on ice, aliquoted, and frozen at −80 °C. Plasma sample aliquots (120 μl) were thawed on ice, and centrifuged at 2000×g for 10 min at 4 °C to remove cellular debris, and 100 μl supernatants were transferred into fresh tubes with 1 ml TRIzol-LS (Life Technologies, cat. no. 10296028). 200 μl of chloroform was added to the TRIzol sample, vortex mixed, and spun at 16,000×g for 15 min at 4 °C. Supernatants were transferred into fresh tubes with 1 μl of glycogen (10 mg/ml stock, Roche, cat. no. 1091393001). 500 μl of isopropanol was added, mixed, and incubated for 10 min at room temperature. RNA pellets were recovered following centrifugation at 16,000×g for 15 min and the removal of supernatants. Pellets were washed twice with 80% ethanol. RNA was resuspended in 8 μl of water. RNA concentrations were measured using the Qubit RNA HS Assay Kit (Invitrogen, cat. no. Q32852). Mouse plasma RNA was purified using the same method as human plasma RNA, except that, after the removal of cell debris, 50 μl of plasma aliquots were resuspended with 750 μl of TRIzol-LS and 200 μl of water.

### DNA purification

Cell debris was removed from the 120 μl plasma aliquot as described for the RNA purification, and 150 μl of SDS-Lysis Buffer (50 mM Tris−HCl pH 8.0, 10 mM EDTA, 1% SDS), 250 μl of TE Buffer (10 mM Tris−HCl pH 7.5, 1 mM EDTA), and 10 μl of Pronase stock (Pronase from Streptomyces griseus, Roche Diagnostics, cat. no. 10165913103, dissolved in water at 20 mg/ml) were added and incubated at 42 °C for 1 h. To extract DNA, 500 μl of Phenol-CHCl3-Isoamylalchol (Wako, cat. no. 311-90151) was added and vortex mixed. The mixture was spun for 5 min at 16,000×g. The supernatant was transferred into a fresh 1.5 ml tube with 15 μl of 5 M NaCl and 1 μl of glycogen (10 mg/ml stock, Roche, cat. no. 1091393001). 1 ml of ethanol was added, mixed, and incubated for 30 min at room temperature. The DNA pellet was recovered by centrifugation at 16,000×g for 30 min. The supernatant was removed, and the pellet was washed with 80% ethanol. DNA was resuspended in 20 μl of water. The DNA concentration was measured by the Qubit 1X dsDNA HS Assay Kit (Invitrogen, cat. no. Q33230). The quality of the purified RNA was examined by the RNA 6000 Pico Kit (Agilent Technologies, cat. no. 5067-1513).

### Mitochondrial DNA quantification and statistical analysis

As described previously[78], mitochondrial DNA and host cell genome copy numbers were quantified using the following primer pairs: MT-ND1 gene, ND1 forward, 5′-CCCTAAAACCCGCCACATCT-3′; ND1 reverse, 5′-GAGCGATGGTGAGAGCTAAGGT-3′; human globulin gene, HGB-1 forward, 5′-GTGCACCTGACTCCTGAGGAGA-3′; and HGB-2 reverse, 5′-CCTTGATACCAACCTGCCCAG-3′. Relative copy numbers against pre-flight means were estimated by the differences in the cycle threshold values from the same volumes of purified DNA. Relative quantification values were plotted, and statistical analysis was performed using the GraphPad Prism 9.3.1 software.

### RNA sequencing

Each sequencing library preparation used 2.5 μl of purified RNA. For the human plasma RNA and CD36-enriched RNA fractions, the SMART-seq Stranded Kit (Takara Bio, cat. no. 634443) was used at a half-volume scale following the user manual. The template amount was verified before the second PCR step by performing real-time PCR with

the Fast SYBR Green Master Mix (Thermo Fisher Scientific, cat. no. 4385612) on a 1 μl aliquot of library reaction in order to adjust the PCR cycle for library amplification. PCR products were purified and eluted in 16 μl of Stranded Elution Buffer diluted with water at a 1:1 ratio. Mouse plasma RNA sequencing using the NEBNext Small RNA Library Kit (New England Biolabs, cat. no. E7330S) and 2 × 36 base paired-end sequencing with NextSeq500 (Illumina) were performed as described previously[79].

### Sequencing data analysis

Reads in FASTQ files were imported into CLC Genomics Workbench (CLC-GW, ver.10.1.1, Qiagen), mapped to human (hg19) or mouse (mm10) reference genomes, and quantified using a 57,773-gene (human) or 49,585-gene (mouse) annotation set (downloaded from the CLC-GW server) to obtain the total count values, which were combined into a table. To plot normalized expression values, total counts were normalized by the scaling option in CLC-GW (normalization value = mean, reference = median mean, trimming 5%). Normalized total count values were log-2 transformed after adding a pseudocount of unity. ANOVAs and Empirical analyses of DRRs as described in figure legends were performed in CLC-GW.

### DNA sequencing

DNA was purified from input plasma and the CD36 fraction was sonicated with Bioruptor II (Sonic Bio) on the high setting for 10 min and converted to sequencing libraries using the NEBNext Ultra II DNA Library Prep Kit (New England Biolabs, cat no. E7645) at half-scale. Library products with insert sizes smaller than ~500 bp were selected using AMPure XP beads (Backman Coulter, cat. no. A63882). According to our examination, the majority of mtDNA were included in high-molecular-weight (>500 bp insert size) fractions in both total plasma cfDNA and DNA purified from the CD36 fraction. 2 × 36 base paired-end sequencing was performed, and FASTQ files were imported and mapped to the hg19 human reference genome by CLC-GW.

### Visualization of mapped RNA-seq and DNA-seq reads

All the mapped RNA-seq reads for each sample were combined into pre-, in-, and post-flight groups. Due to variations in the data quantity and sample numbers in each phase, we decided to use all the reads for visualization. Mapped RNA-seq and DNA-seq reads were exported from CLC-GW as BAM format files, converted into bedGraph format files, and the sections corresponding to the mitochondrial genome were then selected and uploaded to the UCSC Genome Browser (https://genome.ucsc.edu/).

### Correlation analysis

Normalized quantification values of bulk cfRNA-seq for 466 genes from 64 samples (six astronauts, 11 time points, two samples removed) were imported to GraphPad Prism 9.3.1. A Pearson correlation matrix was created, and results were imported into Morpheus to draw a heatmap. Gene annotations for pathways and tissue terms were obtained from the Metascape output, including all genes in each summary category.

### Antibody screening and immunopurification of plasma fraction

An antibody panel of 361 surface markers (LEGENDScreen Human PE Kit, BioLegend, cat. no. 700007, lot no. B328253) was used. Each antibody was reconstituted by adding 25 μl of water. For screening, 1 μl of each antibody was pooled. For the first round, 48 wells from rows A to H of four plates were pooled. For the second round, columns 1–6 and 8–12 of row G were combined into 6 pools. For the third round, antibodies from columns 2 and 8 in row G from four plates were used individually. For each round of screening, 120 μl of plasma aliquot was diluted with 750 μl of PBS, and 100 μl of diluted plasma was combined with pooled antibodies. The mixture was

rotated at 4 °C for 1 h. At this incubation step, the dilution of each antibody was 1:100. For each immunopurification, 20 µl of Dyna-beads Protein G (Invitrogen, cat. no. 1004D) were washed twice with PBS and resuspended in 30 µl of PBS with 1 µl of monoclonal goat anti-mouse IgM (Abcam, cat. no. ab9167, lot no. GR125217-5) and 0.1 mg/ml of BSA (New England Biolabs, B9001S). Washed Protein G beads were added to the plasma samples with antibodies and incubated at 4 °C for 30 min. At this incubation step, the dilution of the IgM antibody combined with Protein G beads was 1:130. After binding, beads were washed twice with ice-cold PBS. Then, 25 µl of SDS–Lysis buffer, 25 µl of TE buffer, and 5 µl of pronase stock were added to the beads and incubated at 42 °C for 30 min. Eluted DNA was combined with 150 µl of TE. DNA was then purified following the plasma DNA purification protocol at half-scale. Antibody clone information and plate layout are provided by the manufacturer at https://www.biolegend.com/Files/Images/media_assets/pro_detail/datasheets/700011_Hu_LEGENDScreen_Layouts.xlsx.

### Purification of CD36 plasma fraction and RNA extraction

The 120 µl plasma aliquot was centrifuged to remove cell debris. The 100 µl supernatant was transferred into a fresh tube and combined with 650 µl of PBS, and 125 µl of the result was kept as input. The rest of the diluted plasma was combined with a diluted anti-CD36 antibody (2 µl of antibody and 40 µl of PBS with 1 × BSA per purification). The mixture was rotated at 4 °C for 1 h. Then, 20 µl of washed protein G beads combined with the goat anti-IgM antibody was added and rotated at 4 °C for 1 h. The supernatant was removed, and the beads were washed twice. Next, 1 ml of 1 × TRIzol (diluted TRIzol-LS with $H_2O$ at 3:1 for 1× concentration) was added to the washed beads. 850 µl of TRIzol-LS was added to the input. RNA was purified following the plasma RNA purification protocol.

### Pathway and tissue-specificity analysis

Metascape 3.5 (https://metascape.org/gp/index.html) analysis was performed on filtered gene lists using default settings. Mouse genes were imported as mouse genes and analyzed as human genes. DAVID 2021 (https://david.ncifcrf.gov/tools.jsp) analysis was performed using default settings. Human gene expression profiles were obtained from the GTEx Portal (https://gtexportal.org/home) with the file that contains median gene-level transcript per million (TPM) by tissue "GTEx_Analysis_2017-06-05_v8_RNASeQCv1.1.9_gene_tpm.gct.gz". Genes that matched within the filtered gene lists were selected for clustering and visualization.

### Plots, heatmaps, and visualization

PCA plots, scatterplots, and normalized value plots were produced using the GraphPad Prism 9.3.1 software. Metascape and DAVID analysis output values and pathway terms were visualized using the bubble plot option in GraphPad Prism 9.3.1. Heatmaps were created using Morpheus (https://software.broadinstitute.org/morpheus/). Normalized log-2 total count values for filtered genes were exported from CLC-GW to Morpheus in Excel format. Clustering trees were produced using the One-minus Pearson option in the Morpheus clustering software.

### DisGeNET analysis

DisGeNET human disease term enrichment results were obtained from Metascape. Lists of DRRs (human, pre-, in-, post-flight), 215 DRRs (mouse, AG vs. MG), and 162 DRRs (mouse, MG vs. GC) were uploaded to the Metascape webtool and analyzed as human genes by default settings. DisGeNET[80] result files were downloaded, and human disease terms were filtered by Log$P < -4$. The Venn diagram in Fig. 4f was produced using Venny 2.1 (https://bioinfogp.cnb.csic.es/tools/venny/index.html). Overlapping parts of the Venn diagram (total 79 terms)

were depicted in Fig. 4g using −Log$P$ (circle size) and gene counts (color) after sorting by overlap group and Log$P$ values using the bubble plot option in GraphPad Prism 9.3.1.

## Data availability

The processed files for human RNA-seq data generated in this study have been deposited in the GeneLab database under accession code OSD-530. Mouse raw and processed RNA-seq data sets have been deposited in the GeneLab database with accession code OSD-532 and in the Gene Expression Omnibus database under accession code GSE213808. Since astronauts who belong to space agencies are highly identifiable persons, individual medical and flight-related records were retained by personal information managers and not released to the researchers who performed sample processing and data analysis. RNA-seq raw datasets, which contain genome sequence information, are not publicly available according to the approved conditions of this study. Data access requires ethics committee approvals for the addition of researchers to the study plan. MM is responsible for explaining further details about the ethical circumstances and controlled access for RNA-seq raw data in response to requests within 4 weeks.

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

## Acknowledgements

We thank the Japan Space Forum for project management, and N. Inoue, K. Murakami, and M. Shirakawa for advice. This study was supported by JAXA, NASA, and JSPS KAKENHI JP20H03234, JP23H02458. The MHU-1 experiment, which provided mouse plasma samples, was supported by JAXA and 14YPTK-005512. Figures 1a and 6a were created with images from the NASA Image Gallery.

## Author contributions

M.M., S.F., S.T., T.K., D.S., and T.A. designed and supervised the study. M.M. performed sample processing and data collection. M.M., N.H., S.-I.F., and Y.S. analyzed data and discussed relevant literature to develop results and discussion sections. M.M. wrote the manuscript. All authors viewed and approved the manuscript before submission.

## Competing interests

The authors declare no competing interests.
