## [Peer Review File · Nature Communications]

Release of CD36-associated cell-free mitochondrial DNA and RNA as a hallmark of space environment responseREVIEWER COMMENTS

Reviewer #1 (Remarks to the Author):

The authors analysed in detail cell-free cfRNA and cfDNA in the plasma of six astronauts before, during and after the flight. Similar analyses were also carried out on samples from mouse spaceflight experiments, ground-based controls, orbital 1G controls and microgravity. The results showed that in all cases exMT DNA and RNA increased under microgravity and were highly reproducible. They were also shown to be enriched in extracellular vesicles, which were found to be the only vesicles with CD-36 from the 361 antibody panel.

In the last few years, NASA twin studies and data collected by 14 astronauts on previous short missions have led to the first studies examining cfDNA in astronauts. In addition, the central biological effects of spaceflight have also been investigated by multi-omics analysis (Cell 2020). All of these studies have highlighted the emerging finding that mitochondrial stress is a consistent phenotype in astronauts. The present study is consistent with these results, and in addition to the analysis of damaged cell-derived cfRNAs, further results clearly show similarities with spaceflight in mice.

The development of a technique for the comprehensive analysis of the biological effects of spaceflight using plasma components presented in this paper will be extremely useful in various medical diagnostic fields in the future, and the significance of the results of this research is judged to be very high.

On the other hand, we would like to request that the following points are added or amended in the explanation so that the readers of this paper can understand them more clearly.

Major comments:

The authors have found a predominant increase in mucin RNAs and neuromuscular-derived RNAs among post-flight cfRNAs. On the other hand, there is little discussion of the factors that led to these results. Possible matters, such as changes in diet between orbit and return, and the effects of rehabilitative exercise under gravity, need to be considered.

Minor comments:

1. "Total RNA and DNA" in line 110 should be "Total cfRNA and cfDNA".
2. In line 112, "measure chromosomal and mitochondrial DNA" should be "measure ratio of chromosomal and mitochondrial DNA in total cfDNA".
3. In line 116 et seq, "RNA" should be "cfRNA".
4. In line 118, "Two samples were" should be "Two samples (each pre- and post-flight one) were". The

reason for this should also be stated briefly. For example, unexpected decomposition, etc?

5. Many of the technical terms used in the text are terms commonly used in transcriptome and genetic analysis. For example, "mitochondrial DNA copy number (line 113)" and "difference in regulated genes (DRG) (line 124)". On the other hand, this study discusses differences in the content of DNA and RNA molecules found as components of cfDNA and cfRNA and does not discuss direct differences in replication or transcription levels. Therefore, the use of terminology used in normal transcriptome analysis. may be misleading to the reader and needs to be improved. Similarly, line 113, for 'individual genes', would 'individual cfRNA molecules' be better? Lines 154 and 155: "expression" may be "levels of cfRNA" or "content of cfRNA"

6. Lines 130, 132: "RNAs for" may be "cfRNAs from", Line 145: "Time course profiles of genes" to "Time-course profiles of cfRNAs" Line 159 "Genes" may be "cfRNAs"

Reviewer #2 (Remarks to the Author):

Overall, this is an important study that analyses human plasma cell-free RNA collected before, during, and after spaceflight and supports previously reported mitochondria dysregulation in space. The study suggests that plasma cfRNA can capture longitudinal gene expression responses of internal tissues to the spaceflight environment. Although the idea of plasma RNA biomarkers reflecting environmental impacts on body conditions is not new and was substantiated by prior publications, another validation that the samples could be collected for such analysis in space is valuable. However, the specific goals of the study were not clearly defined; the study is descriptive and may therefore be of limited interest to a broad readership.

I have the following comments and questions:

1. Dysregulated genes (DRGs) were determined between in-flight and post-flight. It would also be important to determine DRGs from pre-flight to post-flight to highlight and further investigate those not exhibiting complete recovery to normal levels after the flight. Such analysis might lead to additional critical insights.

2. "We found that a broad range of tissues release exMT-containing EVs into the plasma during spaceflight, which may indicate systemic metabolic stress responses in the human body under microgravity."

The data about the tissue-specific DRGs is limited and not entirely convincing. Generally, there was substantial prior research aiming to discover human brain EVs in plasma, but no reliable markers have been identified thus far. The listed "tissue-specific" genes, such as "cerebellum-related" genes CACNA1A and PAX6, albeit enriched in the brain, are also expressed in other tissues. To better validate this point, a

more detailed qRT-PCR-based analysis of individual truly tissue-specific transcripts should be included.

3. Screening with antibodies for surface markers identified an extracellular mitochondria (exMT)-enriched fraction associated with the scavenger receptor CD36. Through three rounds of antibody screenings, only one antibody, the anti-CD36 antibody, consistently enriched mtDNA.

Since the screen was performed using a single mitochondrially encoded gene, ND-1, the question remains whether CD36 pulldown enriches for mitochondria and mitochondrial DNA and RNA, not just for the ND-1 gene. From the most enriched RNAs captured by anti-CD36 (Fig. 3e), what and how many are characteristic of mitochondrial RNome? What about the mitochondrial genes/ DNA?

4. The study could have suggested specific biomarkers for spaceflight-associated conditions such as neuro-ocular syndrome. However, it falls short of identifying them. Are any specific disease-associated genes, e.g., RPE65, been validated as differentially expressed in pre-flight/in-flight conditions? Is it possible to suggest associations with health issues in astronauts (even considering a small sample size)?

5. The writing, and especially the title and abstract, should be revised for accuracy. "CD36-associated extracellular mitochondria" in the title appears inaccurate as it implies the release of entire organelle rather than mtDNA/RNA fragments. EVs are mentioned across the paper, but statements such as "These results indicated that exMT released during spaceflight is associated with CD36-marked EVs" are not supported by the data.

Reviewer #3 (Remarks to the Author):

The manuscript by Ruter et al investigated the effect of space travel on six astronauts. By analyzing RNAs purified from plasma samples collected at 11 different time points before, during, and after the flight, the authors found elevated expression of mitochondrial RNAs and DNAs during spaceflight. They then used a panel of 361 antibodies and found that these mtRNAs were associated with EVs with CD36 as a surface molecular marker. Moreover, the origins of these CD36-expressing EVs were estimated by using RNA-seq and a tissue specificity analysis using DAVID. Lastly, the results were compared with blood samples from mice that underwent 2016 JAXA mission to the ISS, which also showed an elevation of genes encoded by the mitochondrial genome.

Specific comments:

1) It is unclear whether the mtRNAs that authors detected are inside mitochondria or released outside the mitochondria due to stress. There are numerous reports that indicate cytosolic and even extracellular release of mtRNAs during stress.

2) Is there any cell-free mtRNAs that are not inside EVs?

3) The authors listed many potential tissue origins of EVs, such as the brain, muscles, retina, etc. However, it is unclear whether these tissues are already known to release EVs or the release of EVs is flight-induced. Perhaps, authors could compare the amount of EVs released during spaceflight.

4) Significance of elevated mtRNAs is also unclear. Is it just a stress response or does it have some pathological implications?

5) In general, statements in the discussion are too strong. For example, increased mtRNA release from eye-related origins may not necessarily be linked to SANS.

6) In addition, it is also too strong to relate the release of exMT-containing EVs to mitochondrial dysfunction during spaceflight.

REVIEWER COMMENTS

Reviewer #1 (Remarks to the Author):

The authors analysed in detail cell-free cfRNA and cfDNA in the plasma of six astronauts before, during and after the flight. Similar analyses were also carried out on samples from mouse spaceflight experiments, ground-based controls, orbital 1G controls and microgravity. The results showed that in all cases exMT DNA and RNA increased under microgravity and were highly reproducible. They were also shown to be enriched in extracellular vesicles, which were found to be the only vesicles with CD-36 from the 361 antibody panel.

In the last few years, NASA twin studies and data collected by 14 astronauts on previous short missions have led to the first studies examining cfDNA in astronauts. In addition, the central biological effects of spaceflight have also been investigated by multi-omics analysis (Cell 2020). All of these studies have highlighted the emerging finding that mitochondrial stress is a consistent phenotype in astronauts. The present study is consistent with these results, and in addition to the analysis of damaged cell-derived cfRNAs, further results clearly show similarities with spaceflight in mice.

The development of a technique for the comprehensive analysis of the biological effects of spaceflight using plasma components presented in this paper will be extremely useful in various medical diagnostic fields in the future, and the significance of the results of this research is judged to be very high.

On the other hand, we would like to request that the following points are added or amended in the explanation so that the readers of this paper can understand them more clearly.

We would like to thank the reviewer for supporting our approach to cell-free RNA analysis. To assess changes in the post-flight cfRNA samples, we added pairwise comparison results and discussions. We appreciate the comments highlighting the issues on technical terms referring to plasma cell-free components. We agree with the reviewer and have amended the text throughout the revised manuscript. In response to each point, we revised the manuscript as follows:

Major comments:

The authors have found a predominant increase in mucin RNAs and neuromuscular-derived RNAs among post-flight cfRNAs. On the other hand, there is little discussion of the factors that led to these results. Possible matters, such as changes in diet between orbit and return, and the effects of

rehabilitative exercise under gravity, need to be considered.

We thank the reviewer for pointing out these issues. We agree that the description of and discussion about the increase in mucin and tissue-related RNAs in the plasma cfRNA were weak in the original manuscript. We now clarified some technical problems and discussed these points with our interpretation in two aspects.

(i) Difficulties in defining baseline due to variations between the six astronauts

Due to relatively large variations between the six astronauts for the three pre-flight time points, as shown in Extended data fig. 3b, we could not confidently define baseline levels of cfRNAs. For this reason, we were not able to conclude if the changes in mucin and neuro-muscular-related cfRNAs during transition from in-flight to post-flight phases represented an increase from baseline or a return to baseline. This situation is now explained more clearly by providing Extended data fig. 4, showing that only one RNA can be filtered as a differentially represented RNA (DRR) by standard pairwise comparison (FDR-adjusted-p-value < 0.05, |fold change (fc)| > 2) between pre- versus post-flight groups. Since this issue is caused by pre-flight baseline samples, comparisons using individual time points were also not valid when we filter DRRs using relatively conservative criteria of FDR-adjusted-p-value < 0.05 and |fc| > 2. In the revised manuscript, we added a description about this variation issue (lines 124-126) and discussed mucin and tissue-related cfRNA changes mainly based on the idea that these RNAs showed significant reduction during spaceflight and returned to a pre-flight level in the post-flight phase (lines 383-408).

It is possible to analyze the data in detail to examine potential transient increases in cfRNAs by comparing each time point in the post-flight phase. However, an analysis with a small sample size (N=6) of clinical samples may not be conclusive. For potential re-analysis by the research community with less stringent criterion, mean quantification values, SEM values, and results from statistical analyses were uploaded to the GeneLab database.

(ii) Interpretation of spaceflight-associated changes in mucin and tissue-related cfRNAs

Following the reviewer's comment, we aimed to clarify if changes in mucin and other tissue-related cfRNAs share similar mechanisms. The correlation analysis of cfRNAs shown in Extended data fig. 5 indicated that the majority of differentially represented cfRNAs, including mucin genes, were regulated similarly, even these mucin cfRNAs were expected to have different tissue specificities. This supported the idea that various internal tissue responses were caused by common mechanisms, and cfRNA profiles may be complex mixtures of a variety of regulatory statuses. To propose potential explanation to these observations, we added a discussion about potential links to systemic regulation

of blood-tissue barrier functions (lines 383-408), which were also proposed by recent mouse experiments¹⁻⁵.

According to our IRB-approved research plan, which was one of the first few cases of genomic analysis of astronaut participants, we did not intend to perform analyses on individual variation and clinical information. Therefore, in this study, we did not have access to astronauts' activity logs or information regarding their health conditions. With our current study showing the technical feasibility of liquid biopsy-based omics analyses and the necessity of detailed analyses, we hope to justify personal data access in any future studies towards personalized space medicine and individualized countermeasure development based on omics analysis.

Minor comments:

1. *"Total RNA and DNA" in line 110 should be "Total cfRNA and cfDNA".*

We thank the reviewer for pointing this out. Correction is made in the text (line 112).

2. *In line 112, "measure chromosomal and mitochondrial DNA" should be "measure ratio of chromosomal and mitochondrial DNA in total cfDNA".*

We thank the reviewer for pointing this out. Correction is made in the text (line 114).

3. *In line 116 et seq, "RNA" should be "cfRNA".*

We thank the reviewer for pointing this out. Correction is made in the text (line 119).

4. *In line 118, "Two samples were" should be "Two samples (each pre- and post-flight one) were". The reason for this should also be stated briefly. For example, unexpected decomposition, etc?*

We thank the reviewer for the comment. We corrected the sentence and explained the situation – two samples were removed due to technical problems seen during sequencing library preparation that indicated poor RNA quality or yield (lines 121-123).

5. *Many of the technical terms used in the text are terms commonly used in transcriptome and genetic analysis. For example, "mitochondrial DNA copy number (line 113)" and "difference in regulated genes (DRG) (line 124)". On the other hand, this study discusses differences in the content of DNA and RNA molecules found as components of cfDNA and cfRNA and does not discuss direct differences in replication or transcription levels. Therefore, the use of terminology used in normal transcriptome analysis. may be misleading to the reader and needs to be improved. Similarly, line 113, for 'individual genes', would 'individual cfRNA molecules' be better? Lines 154 and 155: "expression" may be "levels of cfRNA" or "content of cfRNA"*

We thank the reviewer for pointing this out. We agree about the misleading terms in the original manuscript, and have made corrections throughout the revised manuscript.

6. Lines 130, 132: "RNAs for" may be "cfRNAs from", Line 145: "Time course profiles of genes" to "Time-course profiles of cfRNAs" Line 159 "Genes" may be "cfRNAs"

We thank the reviewer for pointing out these words. We made changes throughout the revised manuscript accordingly.

Reviewer #2 (Remarks to the Author):

Overall, this is an important study that analyses human plasma cell-free RNA collected before, during, and after spaceflight and supports previously reported mitochondria dysregulation in space. The study suggests that plasma cfRNA can capture longitudinal gene expression responses of internal tissues to the spaceflight environment. Although the idea of plasma RNA biomarkers reflecting environmental impacts on body conditions is not new and was substantiated by prior publications, another validation that the samples could be collected for such analysis in space is valuable. However, the specific goals of the study were not clearly defined; the study is descriptive and may therefore be of limited interest to a broad readership.

We thank the reviewer for supporting the value of the cfRNA analysis in space research. It is valid criticism that, compared to plasma cfDNA and cfRNA studies in other fields, the initial goals of the study were unclear. In this manuscript, we intended to describe our initial analysis of astronaut samples and to describe study details for sharing processed datasets in the GeneLab database. We particularly appreciated the reviewer's helpful comments that pointed out distinctions between exMT and other types of mitochondrial contents release, and that encouraged us to perform further analysis on the CD36 fraction. Although we could not perform all the requested data collection, we added new figures and descriptions in the results and discussion sections.

I have the following comments and questions:

1. Dysregulated genes (DRGs) were determined between in-flight and post-flight. It would also be important to determine DRGs from pre-flight to post-flight to highlight and further investigate those not exhibiting complete recovery to normal levels after the flight. Such analysis might lead to additional critical insights.

We thank the reviewer for pointing out the significance of pre-flight vs post-flight comparisons. We agree that the difference between the pre-flight baseline and post-flight time points should highlight if any part of spaceflight associated changes were fully recovered to baseline levels or not after post-flight 120 days. However, we did not draw any specific conclusions in the original manuscript because of the relatively large variation among 6 astronauts and 3 repeats of baseline measurements. To clarify this point, we now included an additional description related to Fig. 1 (lines 124-126) and Supplementary data fig. 4. We would like to highlight that clinical sample analysis with N=6 sampling is usually not conclusive. However, we provided processed data in the GeneLab database for the research community to examine how each cfRNA changed through the time-course sampling. Please refer to our response to the Major Point from Reviewer #1.

2. "We found that a broad range of tissues release exMT-containing EVs into the plasma during spaceflight, which may indicate systemic metabolic stress responses in the human body under microgravity."

The data about the tissue-specific DRGs is limited and not entirely convincing. Generally, there was substantial prior research aiming to discover human brain EVs in plasma, but no reliable markers have been identified thus far. The listed "tissue-specific" genes, such as "cerebellum-related" genes CACNA1A and PAX6, albeit enriched in the brain, are also expressed in other tissues. To better validate this point, a more detailed qRT-PCR-based analysis of individual truly tissue-specific transcripts should be included.

We agree with the reviewer's comment. Gene lists derived from pathway analysis indicate biological processes but not tissue specificity. There are gene expression profiles generated at single cell resolutions for major tissue types, including the whole brain, cerebellum, and spinal cord for human and mouse^{6,7}. Other cell marker databases, such as PangaoDB (<https://panglaodb.se/index.html>) and CellMarker2.0 (<http://bio-bigdata.hrbmu.edu.cn/CellMarker/index.html>), list much larger numbers of genes, using less stringent criteria, as cell-type specific markers. However, these examples are still not sufficient to exclude minor contributions from multiple cell types due to tissue heterogeneity. For example, neuron specific genes are expected to be expressed in peripheral tissues through projections of neurons. Considering such challenges in identifying truly tissue-specific genes, we referred to two human databases, HPA and GTEx, in our revised manuscript. We identified the overlap between tissue-restricted or tissue-enriched genes from HPA and the 466 DRRs derived from the Fig. 1 analysis. We further verified expression profiles in tissue sets in the GTEx database. Through these steps, we selected candidate tissue-specific genes verified with our best effort and presented them in our new Fig. 3 (lines 180-197). Our RNA samples purified from plasma were highly degraded and

their quantity was extremely limited. Therefore, we could not perform RT-PCR verification. However, we provided genome browser views of mapped RNA-seq reads to support the detection of the candidate tissue-specific cfRNAs.

3. Screening with antibodies for surface markers identified an extracellular mitochondria (exMT)-enriched fraction associated with the scavenger receptor CD36. Through three rounds of antibody screenings, only one antibody, the anti-CD36 antibody, consistently enriched mtDNA.

Since the screen was performed using a single mitochondrially encoded gene, ND-1, the question remains whether CD36 pulldown enriches for mitochondria and mitochondrial DNA and RNA, not just for the ND-1 gene. From the most enriched RNAs captured by anti-CD36 (Fig. 3e), what and how many are characteristic of mitochondrial RNome? What about the mitochondrial genes/ DNA?

We thank the reviewer for encouraging us to perform additional analyses on mtDNA and mtRNA in the CD36 fraction. Following the comment, we performed sequencing analyses of DNA from the input (total plasma cfDNA) and the CD36 fraction in two in-flight samples. The results supported the conclusion that entire mitochondrial DNA was included in the CD36 fraction, with similar representation between the input and the CD36 fraction, except for the D-loop region. These results are now included in the new Fig. 5a and described in the text in lines 251-281.

Similarly, mtRNA representations were compared between bulk cfRNA and the CD36 fraction. RNAs from all the 13 mitochondrially encoded OXPHOS complex subunits were included in the CD36 fraction, but the ratios among the mtRNAs between input cfRNA and the RNA purified from the CD36 fraction were different. Although the mechanisms were still unclear, we discussed the possibility that the CD36 fraction may be derived from a qualitatively different population of mitochondria responding to space environment.

Altered regulatory statuses in the D-loop region by DNA methylation⁸ and DNA packaging⁹ have been described by other studies. Differential regulation of the MT-ND6 gene at the transcriptional¹⁰ and post-transcriptional^{11,12} levels has also been reported. Although the mechanisms and biological significance are unknown, our observations contribute to the field by providing another example of mitochondrial regulation.

4. The study could have suggested specific biomarkers for spaceflight-associated conditions such as neuro-ocular syndrome. However, it falls short of identifying them. Are any specific disease-associated genes, e.g., RPE65, been validated as differentially expressed in pre-flight/in-flight conditions? Is it

possible to suggest associations with health issues in astronauts (even considering a small sample size)?

We thank the reviewer for raising this point. We explored mouse space experiment datasets to identify potential conserved biomarker genes. However, as described in our Fig. 4 results, the overlap between mouse and human results at the gene name level was consistently poor. Instead, we found that pathway level comparisons provided overlap across multiple aspects, including mitochondria, OXPHOS regulation, ROS response, and inflammation as biological processes, and broad tissue types, including the brain, retina, and muscle as tissue terms. Thus, our results may support the advantages of omics-based evaluation of environmental responses to the conventional single-gene biomarkers in personalized space medicine.

RPE65 is indeed one of the important candidates as a biomarker to potentially indicate stress response or cell damage in the retina, and RPE65 was included in the 466 DRRs. Related to the tissue specificity issue, the GTEx database indicated that the RNA expression for RPE65 has been detected in multiple parts of the brain, and, thus, RPE65 was excluded from our strict criteria of tissue-specific cfRNAs. Changes in cfRNA may represent altered tissue contribution as well as RNA expression changes in internal tissues, but our current study has limitation in distinguishing these possibilities.

5. The writing, and especially the title and abstract, should be revised for accuracy. "CD36-associated extracellular mitochondria" in the title appears inaccurate as it implies the release of entire organelle rather than mtDNA/RNA fragments. EVs are mentioned across the paper, but statements such as "These results indicated that exMT released during spaceflight is associated with CD36-marked EVs" are not supported by the data.

We thank the reviewer for the comment. This is valid criticism, and we now made corrections throughout the revised manuscript.

Reviewer #3 (Remarks to the Author):

The manuscript by Ruter et al investigated the effect of space travel on six astronauts. By analyzing RNAs purified from plasma samples collected at 11 different time points before, during, and after the flight, the authors found elevated expression of mitochondrial RNAs and DNAs during spaceflight. They then used a panel of 361 antibodies and found that these mtRNAs were associated with EVs with CD36 as a surface molecular marker. Moreover, the origins of these CD36-expressing EVs were estimated by using RNA-seq and a tissue specificity analysis using DAVID. Lastly, the results were

compared with blood samples from mice that underwent 2016 JAXA mission to the ISS, which also showed an elevation of genes encoded by the mitochondrial genome.

Specific comments:

1) It is unclear whether the mtRNAs that authors detected are inside mitochondria or released outside the mitochondria due to stress. There are numerous reports that indicate cytosolic and even extracellular release of mtRNAs during stress.

We thank the reviewer for highlighting this issue. Our current study has limitations in the characterization of cfRNA, and we could not conclude if the mtRNAs that increased during space flight were inside or outside of mitochondria, or released into the plasma. As part of our description about the limitations of this study, we discussed a cfRNA complex in the plasma, cytosolic mitochondrial components as EV contents, and the release of intact mitochondria as possibilities (lines 339-347). According to these limitations, we also modified expressions in our text and title. With limited samples available from astronauts, we would like to establish assay methods including more sophisticated plasma and EV fractionation and electron microscopy analysis for future studies.

2) Is there any cell-free mtRNAs that are not inside EVs?

We thank the reviewer for raising this issue. Related to the previous point, we could not provide substantial evidence to support any presence of cfRNA outside of EVs.

3) The authors listed many potential tissue origins of EVs, such as the brain, muscles, retina, etc. However, it is unclear whether these tissues are already known to release EVs or the release of EVs is flight-induced. Perhaps, authors could compare the amount of EVs released during spaceflight.

We thank the reviewer for raising this issue. To note the general limitations in our cfRNA and cfDNA analysis, we added references describing the impact of sample handling on mtDNA. As RNA-seq library preparation included amplification steps, it is hard to estimate the absolute cfRNA amount/copy numbers. The use of spike-in controls was also avoided in this study as the amount of purified cfRNA was extremely low.

4) Significance of elevated mtRNAs is also unclear. Is it just a stress response or does it have some pathological implications?

Our current study only measured cfRNA by RNA-seq, and we would like to perform detailed functional

studies in the future. In our revised manuscript, we discussed potential pathological implications with references to other studies. As potential upstream events of mitochondrial regulation, we mainly discussed the potential link between the blood-tissue barrier and tissue inflammation (lines 433-437).

5) In general, statements in the discussion are too strong. For example, increased mtRNA release from eye-related origins may not necessarily be linked to SANS.

We thank the reviewer for pointing out this issue. Since the gene list and associated diseases indicate a link to retina tissue, we referred to SANS as a prominent health issue related to the eye. We now added more descriptions to support the potential link between mitochondrial oxidative stress, neuro-inflammation, and SANS with references (lines 394-408, 433-437).

6) In addition, it is also too strong to relate the release of exMT-containing EVs to mitochondrial dysfunction during spaceflight.

We thank the reviewer for this comment. As pointed out also by Reviewer #2 in the point 5, we made extensive revisions in our manuscript, including the title.

References

- 1 Mao, X. W. *et al.* Impact of Spaceflight and Artificial Gravity on the Mouse Retina: Biochemical and Proteomic Analysis. *Int J Mol Sci* **19** (2018). <https://doi.org:10.3390/ijms19092546>
- 2 Mao, X. W. *et al.* Spaceflight induces oxidative damage to blood-brain barrier integrity in a mouse model. *FASEB J* **34**, 15516-15530 (2020). <https://doi.org:10.1096/fj.202001754R>
- 3 Arima, Y. *et al.* Regional neural activation defines a gateway for autoreactive T cells to cross the blood-brain barrier. *Cell* **148**, 447-457 (2012). <https://doi.org:10.1016/j.cell.2012.01.022>
- 4 Stofkova, A. *et al.* Photopic light-mediated down-regulation of local alpha(1A)-adrenergic signaling protects blood-retina barrier in experimental autoimmune uveoretinitis. *Sci Rep* **9**, 2353 (2019). <https://doi.org:10.1038/s41598-019-38895-y>
- 5 Tanaka, Y., Arima, Y., Kamimura, D. & Murakami, M. The Gateway Reflex, a Novel Neuro-Immune Interaction for the Regulation of Regional Vessels. *Front Immunol* **8**, 1321 (2017). <https://doi.org:10.3389/fimmu.2017.01321>
- 6 Kozareva, V. *et al.* A transcriptomic atlas of mouse cerebellar cortex comprehensively defines cell types. *Nature* **598**, 214-219 (2021). <https://doi.org:10.1038/s41586-021-03220-z>
- 7 Yadav, A. *et al.* A cellular taxonomy of the adult human spinal cord. *Neuron* **111**, 328-344 e327 (2023). <https://doi.org:10.1016/j.neuron.2023.01.007>

- 8 Cao, K. *et al.* Hypermethylation of Hepatic Mitochondrial ND6 Provokes Systemic Insulin Resistance. *Adv Sci (Weinh)* **8**, 2004507 (2021). <https://doi.org:10.1002/advs.202004507>
- 9 Isaac, R. S. *et al.* Single-nucleoid architecture reveals heterogeneous packaging of mitochondrial DNA. *bioRxiv*, 2022.2009.2025.509398 (2022). <https://doi.org:10.1101/2022.09.25.509398>
- 10 Jemt, E. *et al.* Regulation of DNA replication at the end of the mitochondrial D-loop involves the helicase TWINKLE and a conserved sequence element. *Nucleic Acids Res* **43**, 9262-9275 (2015). <https://doi.org:10.1093/nar/gkv804>
- 11 Popow, J. *et al.* FASTKD2 is an RNA-binding protein required for mitochondrial RNA processing and translation. *RNA* **21**, 1873-1884 (2015). <https://doi.org:10.1261/rna.052365.115>
- 12 Jourdain, A. A. *et al.* A mitochondria-specific isoform of FASTK is present in mitochondrial RNA granules and regulates gene expression and function. *Cell Rep* **10**, 1110-1121 (2015). <https://doi.org:10.1016/j.celrep.2015.01.063>

REVIEWERS' COMMENTS

Reviewer #1 (Remarks to the Author):

The Rutter et al. paper has been revised correctly on all points raised, and its novelty and logic are clearly demonstrated. This is a landmark achievement that demonstrates that several interesting cfRNAs, including mitochondria-derived cfRNAs and cfDNAs, are altered in space microgravity despite the limited conditions of astronauts' liquid biopsies. The experiments supporting these results were well performed and described, and proper statistics were used; publication in Nature Communications is strongly encouraged.

Reviewer #3 (Remarks to the Author):

Given the limited resources, I'm satisfied the improvement authors made on the revised manuscript.